# Evaluation of Novel Inhibitors of Tryptophan Dioxygenases for Enzyme and Species Selectivity Using Engineered Tumour Cell Lines Expressing Either Murine or Human IDO1 or TDO2

**DOI:** 10.3390/ph15091090

**Published:** 2022-08-31

**Authors:** Sofian M Tijono, Brian D. Palmer, Petr Tomek, Jack U. Flanagan, Kimiora Henare, Swarna Gamage, Lukas Braun, Lai-Ming Ching

**Affiliations:** 1Auckland Cancer Society Research Centre, Faculty of Medical and Health Sciences, School of Medicine, University of Auckland, Private 92019, Auckland 1142, New Zealand; 2Department of Pharmacology and Clinical Pharmacology, Faculty of Medical and Health Sciences, School of Medicine, University of Auckland, Private Bag 92019, Auckland 1142, New Zealand

**Keywords:** tryptophan dioxygenase inhibitors, immune modulation, cancer models

## Abstract

Indoleamine 2, 3-dioxygenase 1 (IDO1) is commonly expressed by cancers as a mechanism for evading the immune system. Preclinical and clinical studies have indicated the potential of combining IDO1 inhibitors with immune therapies for the treatment of cancer, strengthening an interest in the discovery of novel dioxygenase inhibitors for reversing tumour-mediated immune suppression. To facilitate the discovery, development and investigation of novel small molecule inhibitors of IDO1 and its hepatic isozyme tryptophan dioxygenase (TDO2), murine tumour cells were engineered to selectively express either murine or human IDO1 and TDO2 for use as tools to dissect both the species specificity and isoenzyme selectivity of newly discovered inhibitors. Lewis lung carcinoma (LLTC) lines were engineered to express either murine or human IDO1 for use to test species selectivity of the novel inhibitors; in addition, GL261 glioma lines were engineered to express either human IDO1 or human TDO2 and used to test the isoenzyme selectivity of individual inhibitors in cell-based assays. The 20 most potent inhibitors against recombinant human IDO1 enzyme, discovered from a commissioned screening of 40,000 compounds in the Australian WEHI compound library, returned comparable IC_50_ values against murine or human IDO1 in cell-based assays using the LLTC-mIDO1 and LLTC-hIDO1 line, respectively. To test the in vivo activity of the hits, transfected lines were inoculated into syngeneic C57Bl/6 mice. Individual LLTC-hIDO1 tumours showed variable expression of human IDO1 in contrast to GL261-hIDO1 tumours which were homogenous in their IDO1 expression and were subsequently used for in vivo studies. W-0019482, the most potent IDO1 inhibitor identified from cell-based assays, reduced plasma and intratumoural ratios of kynurenine to tryptophan (K:T) and delayed the growth of subcutaneous GL261-hIDO1 tumours in mice. Synthetic modification of W-0019482 generated analogues with dual IDO1/TDO2 inhibitory activity, as well as inhibitors that were selective for either TDO2 or IDO1. These results demonstrate the versatility of W-0019482 as a lead in generating all three subclasses of tryptophan dioxygenase inhibitors which can be applied for investigating the individual roles and interactions between IDO1 and TDO2 in driving cancer-mediated immune suppression.

## 1. Introduction

Immune evasion is a recognised hallmark of cancer, and there is accumulating evidence that tumours adopt multiple ways to evade host immunity [1]. One key mechanism is mediated through increased expression of the tryptophan-catabolising enzyme indoleamine 2, 3-dioxygenase 1 (IDO1) [2] in the tumour. The increased IDO1 activity results in increased conversion of this essential amino acid along the kynurenine metabolic pathway [3]. The consequent depletion of tryptophan [4,5] and the increase in kynurenine and downstream metabolites [6,7], causes activation and differentiation of suppressive T regulatory cells (Tregs) and inactivation of the host’s tumour-specific killer T cells. The immunosuppressive activity of IDO1 was first discovered in studies investigating mechanisms of foetal protection against the maternal immunity during gestation. Blockage of IDO1 expressed at the maternal–foetal interface resulted in the rejection of foetuses bearing foreign paternal tissue antigens in mice [8]. Evidence that tumours co-opt a similar mechanism to suppress the endogenous immunity of the host, followed from demonstrations that the enzyme is expressed by a broad range of clinical tumours and that IDO1 expression by experimental murine tumours prevented their rejection in pre-immunized mice [2]. IDO1-mediated immune suppression has been documented for all major classes of solid [9] and haematological [10] malignancies, although the precise mechanism of action is far from being completely understood. Of greater relevance to the cancer therapy field, is that clear associations between high IDO1 expression and poor patient outcome have been documented for many difficult-to-treat cancers that include glioma [11], melanoma [12], prostate [13], endometrial [14], ovarian [15], breast [16], pancreatic [17], lung [18,19], and colorectal [20,21] cancers. The compelling clinical evidence of elevated IDO1 expression impacting negatively on tumour immunity and patient survival [9,10,11,12,13,14,15,16,17,18,19,20,21] has prompted many groups [22,23,24,25,26,27,28,29], to investigate novel tryptophan dioxygenase inhibitors for restoring tumour immunity. 

Numerous molecules with IDO1 inhibitory activity have been described, and several have entered advanced clinical evaluation. Epacodostat, a hydroxyamidine analogue developed by Incyte Inc [26], was the first IDO1-selective inhibitor to complete Phase III evaluation in combination with Pembrolizumab, an anti-PD1 immune checkpoint blocker from Merck for the treatment of advanced malignant melanoma (ECHO-301/KEYNOTE-252). Disappointingly, the cohort in the arm which received the combination did not show improved progression-free survival, nor increased overall survival compared to the group treated with Pembrolizumab alone [30]. The poor performance of Epacodostat in Phase III dampened the interest to take further IDO1-selective inhibitors into large and costly Phase III trials. It also led to a shift in focus towards targeting tryptophan dioxygenase 2 (TDO2). Expressed predominantly in the liver, TDO2 is responsible for the physiological maintenance of tryptophan homeostasis in the body [31]. Although less frequently expressed than IDO1, TDO2 can also be detected in cancers, particularly brain [32] and hepatic cancers [33]. The question as to whether inhibition of both enzymes would provide synergistic benefits for cancer patients, remains very much an open question and has fuelled recent efforts into the development of dual IDO1/TDO2 inhibitors [34,35] or TDO2-selective inhibitors [36] that could be used in conjunction with IDO1-selective inhibitors. In this communication, we report the establishment and characterisation of cell lines engineered to constitutively express only murine or human IDO1 or TDO2, suitable for expediting the discovery and pharmacological evaluation of selective and dual inhibitors of IDO1 and TDO2.

## 2. Results and Discussion

### 2.1. Establishment of LLTC Lines Constitutively Expressing Either Murine or Human IDO1 for Cell-Based Assays of Tryptophan Dioxygenases 

Our laboratory has previously screened compound libraries for small molecule inhibitors of tryptophan dioxygenases using enzymatic assays [23,24,34]. As tryptophan dioxygenases are cytoplasmic proteins, it is vital following the enzymatic assays, to measure the potency of the inhibitors against the enzyme in cell-based assays. Commonly used cancer cell lines in the laboratory generally require prior induction with cytokines before they express cytoplasmic tryptophan dioxygenases. For example, we screened 49 lines from our Centre’s extensive bank of primary melanoma lines generated from New Zealand melanoma patients [37] and found that none of the NZ Melanoma lines expressed IDO1, IDO2 and TDO2 prior to exposure to human IFNγ. Following 3 days’ exposure to human IFNγ, 42 (86%) of the lines expressed IDO1 whilst 7 lines (14%) remained negative (Appendix A). The melanoma lines were all negative for TDO2 and IDO2 even after induction with IFNγ. Western blots of eight representative lines showing different levels of IDO1 expression between lines after induction is presented in Figure 1A. 

To overcome the need for IFNγ induction, as well as the variability of expression between experiments, we engineered murine tumour cell lines to constitutively express a specific tryptophan dioxygenase for use in cell-based assays. The LLTC lung carcinoma line and the murine GL261 glioblastoma wild-type lines in our laboratory, appear not to express endogenous murine dioxygenases even after IFNγ induction and were selected for the transfections. Murine cell lines were chosen over human cell lines, as murine lines can be implanted into fully immunocompetent hosts for evaluating the immune effects of the inhibitors. A transfected LLTC line that constitutively expressed only the human IDO1 (LLTC-hIDO1), and a line that constitutively expressed only the murine homologue (LLTC-mIDO1) were established for testing of potential species selectivity of inhibitors. Western blotting (Figure 1B) and immunohistochemistry (Figure 1C) analyses confirmed that the wild-type LLTC line (LLTC-WT) do not express endogenous murine IDO1 even with IFNγ induction, and also confirmed that the transfected LLTC-mIDO1 line expressed only the murine homologue, whilst the LLTC-hIDO1 line expressed only the human IDO1 homologue (Figure 1B,C). The LLTC-mIDO1 and the LLTC-hIDO1 transfected lines expressed similar amounts of IDO1 protein with or without IFNγ induction (Figure 1B).

The 20 most potent inhibitors discovered in our commissioned screening of the WEHI library [22] using a recombinant human IDO1 enzymatic assay [24], were evaluated in cell-based assays using the transfected LLTC lines. Each of the ‘hits’ was assayed in technical triplicates of 6 serial dilutions from a top concentration, for inhibition of either murine IDO1 or human IDO1 in the LLTC-mIDO1 and LLTC-hIDO1 lines, respectively. Two known IDO1 inhibitors, 4-phenyl-imidazole (4-PI) and the INCB14943 derivative of Epacodostat [26] were included as reference controls for comparison. When the concentration for 50% inhibition (IC_50_) measured against LLTC-hIDO1 cells was compared to that measured against LLTC-mIDO1 cells, comparable activity against human and murine IDO1 at non-toxic concentrations was shown for all the inhibitors (Figure 1D, Table 1). Species selectivity would not be a problem in the use of these compounds. 

### 2.2. IDO1 Expression and Growth of LLTC-WT and LLTC-hIDO1 Tumours in Mice

The LLTC lines were next evaluated for their suitability for use for evaluating the pharmacology of novel IDO1 inhibitors in mice. LLTC-WT and LLTC-hIDO1 cells were each implanted subcutaneously in syngeneic C57Bl/6 mice. Tumours developed from both lines with wild-type tumours growing at a slightly faster rate than the LLTC-hIDO1 tumours (Figure 2A), contrary to what would be expected. Western blot analysis of protein samples from individual 14-day LLTC-hIDO1 tumours showed variability in intensity in their human IDO1 bands, with some tumours showing the appearance of a doublet (Figure 2B). The basis for the doublet from tumour samples is not clear, as extracts from the cultured LLTC-mIDO1 and LLTC-hIDO1 cells, show a single band of their respective transfected homologue only. We carried out an experiment to determine whether the variability in hIDO1 expression in individual tumours was observed in subsequent serial transplants from a first-generation (T1) high hIDO1-expressing tumour (Figure 2C), with the underlying intent of establishing a stable line of high hIDO1-expressing tumours for in vivo studies. Disappointingly, variable expression of hIDO1 was observed in all three subsequent generations of tumours derived from the initial T1 high-expressor. Intriguingly, implants from a third-generation tumour (T1.1.2) that expressed little or no IDO1 could give rise to high IDO1-expressing tumours (e.g., T1.1.2.3) in the following generation of new implants (Figure 2C). Variability in hIDO1 expression at each generation was obtained whether we transplanted discrete cubes of tumour tissue or aliquots of the homogenised parental tumour (data not shown). The basis for the variability in hIDO1 expression between tumours is not clear and we subsequently focused on the use of tumours developing from transfected GL261 lines for in vivo studies instead.

### 2.3. Growth and IDO1 Expression by GL261-WT and GL261-hIDO1 Tumours in Mice

The murine GL261 glioma line has been widely used as a preclinical model for investigating the potential benefits of dioxygenase inhibitors and immune checkpoint blockades in the treatment of glioblastoma multiforme [11,38,39,40]. The GL261-WT and the transfected GL261-hIDO1 cells were inoculated subcutaneously into C57Bl/6 syngeneic mice. As expected, tumours developing from GL261-hIDO1 cells grew faster than GL261-WT tumours (Figure 3A). Western blots of their protein extracts confirmed that GL261-WT cells and their 14-day tumours express neither murine nor human IDOI. GL261-hIDO1 cells and their tumours expressed only the transfected human IDO1 protein and no endogenous murine IDO1 protein (Figure 3B). Individual 14-day GL261-hIDO1 tumours developing from the same inoculum of GL261-hIDO1 cells all expressed similar amounts of human IDO1 protein (Figure 3B). 

Changes in the ratio of kynurenine to tryptophan concentrations (K:T) in plasma are commonly used as surrogate markers of intratumoural IDO1 activity. In mice implanted with GL261-hIDO1 tumours, the plasma K:T ratios increased steadily with days of tumour implantation (Figure 3C) and correlated with the growth and increasing size of GL261-hIDO1 tumours in the mice (Figure 3D(b)). Significant increases in plasma K:T ratios in plasma of mice implanted with wild-type GL261 tumours were not observed until 21 days after implantation, reflecting the slower tumour growth rate and the low and variable amounts of IDO1 expressed by individual wild-type GL261 tumours (Figure 3D(a)). 

Flow cytometric analysis of immune cell subsets showed that the proportion of CD45^+^ leucocytes with the B220^+^ B-lymphocyte marker in 15-day GL261-WT and GL261-hIDO1 tumours were similar and represented less than 5% of the total leucocytes (Figure 3E). However, numbers of CD3^+^ T-lymphocytes in wild-type tumours were double those detected in GL261-hIDO1 tumours (Figure 3F). Moreover, when the CD3^+^ population was subclassified into CD4^+^ T-helpers, CD8^+^ T-effector, or Foxp3 T-suppressor cells, the wild-type tumours contained a larger proportion of CD8^+^ cells, whilst the hIDO1-expressing tumours had a higher proportion of Foxp3 T-regulatory cells (Figure 3F). These differences in immune cell profiles between the wild-type and IDO1-expressing GL261 lines are consistent with clinical observations that patients with high IDO1-expressing cancers have a suppressed cytotoxic T-cell function and an increased T-regulatory suppressor cell activity [2,3,4,5,6,7,8]. Such observations have provided the rationale for the considered use of IDO1 inhibitors to overcome immune suppression in cancer patients. The GL261-hIDO1 tumours would provide a valuable in vivo model for the evaluation of novel IDO1 inhibitors.

### 2.4. Evaluation of W-0019482 as a Lead for the Synthesis of New IDO1 Inhibitors for Cancer Therapy

W-0019482 emerged as one the most promising ‘hits’ from our screening of the WEHI library for compounds with IDO1 inhibitory activity (Figure 1D). W-0019482 returned an IC_50_ of 80 nM compared to 20 nM for the INCB14934 analogue of Epacodostat from Incyte, both measured using the LLTC-hIDO1 cell-based assay (Table 1). We also compared the IC_50_ values measured in the cell-based assay to those obtained in the cell-free recombinant hIDO1 enzymatic assay for W-0019482 (Figure 4a) and two other clinical leads; NewLink Genetics (Figure 4b), and INCB14934 from Incyte (Figure 4c). Whilst the titration curve for INCB14934 in the cell-assay was very similar to that obtained in the enzymatic assay (Figure 4c), NLG919 was 10-fold more potent in the enzymatic assay compared to the cell-assay (Figure 4b). W-0019482, surprisingly, was 30-fold more potent in the cell-based assay compared to its activity against the enzyme (Figure 4a). We have yet to investigate the basis of the greater inhibitory activity of W-0019482 in cells. It may be due to assisted transport through the plasma membrane leading to high intracellular accumulation. Given that IDO1 is an intracellular protein, this property of W-0019482 would be a definite advantage to have in a therapeutic agent targeting this enzyme.

We measured the ability of W-0019482 to block tryptophan conversion in tumour-bearing mice by monitoring changes in K:T ratios following treatment with a single ip administration. K:T ratios in plasma and tumour were reduced within an hour after W-0019482 injection, with a faster and more pronounced decrease in the K:T ratios in the tumour compared to that in plasma (Figure 5A). K:T ratios in the tumour had returned to approximately 50% of that observed in mice without treatment (Figure 5A). A single daily ip dose of W-0019482 may be sufficient to provide continuous inhibition of tryptophan metabolism and reverse IDO1-induced immune suppression in the tumour-bearing hosts.

We next measured the ability of a daily ip administration of W-0019482 to control the growth of subcutaneous GL261-hIDO1 in immune competent syngeneic C57Bl/6 hosts (Figure 5B). Daily ip administration of W-0019482 at three different doses up to its maximum tolerated dose (MTD) of 150 mg/kg all delayed the growth of subcutaneous GL261-hIDO1 tumours. The growth inhibition observed with a dose of 75- and 100-mg/kg were similar and slightly better than that obtained at the MTD (Figure 5B).

### 2.5. Inhibition of Tryptophan Dioxygenases by Analogues of W-0019482

W-0019482 exhibited many properties desirable as a lead for the development of inhibitors for targeting tryptophan dioxygenases. Derivatives of W-0019482 have been synthesised in-house [22] and we tested a small panel of these for their potency and selectivity for inhibiting IDO1 and TDO2. Although mRNA for TDO2 has been reported to be expressed in the GL261 line [33], we have not been able to detect TDO2 protein expression in the wild-type GL261 line in our laboratory. A GL261 line that constitutively expressed human TDO2 (GL261-hTDO2) was engineered for use for these studies of enzyme-selectivity of W-0019482 and a select panel of its analogues with IC_50_ < 50 µM against human IDO1 in the enzymatic assay. Each compound was tested in cell-based assays using LLTC-hIDO1 and GL261-hTDO2, respectively, for their potency at inhibiting human IDO1 and human TDO2. W-0019482 was initially selected as a hit based on its potency against IDO1. However, the results in Table 2 and Figure 6 establish that synthetic modification of W-0019482 can lead to analogues with either dual or TDO2-selective profiles, as well as to more potent IDO1-selective inhibitors. Four representative examples of compounds exhibiting comparable activity against both cellular hIDO1 and hTDO2 with the ratio of IC_50_ values ≤ 2-fold different from each other are shown in Table 2A and Figure 6A. These compounds we have considered as dual inhibitors and are structurally diverse. Four derivatives exhibiting IC_50_ values for hIDO1 that were 8.1–5.8 fold higher than that measured against hTDO2 (Table 2B, Figure 6B), have been categorised as being TDO2-selective inhibitors. It is interesting to note that introduction of a Cl, Br or I substituent at the 5-position of the isoxazolopyridine core, as the sole substituent on the core, increases TDO2 potency, resulting in TDO2 selectivity. Four other derivatives exhibiting greater activity against cellular hIDO1 than cellular hTDO2 were categorised as IDO1-selective inhibitors (Figure 2C, Table 2C). One of these, SN37313 was even more potent than the parental hit, W-0019482. SN36494 is the 5-bromo analogue of the W-0019482 hit and retains its IDO-1 selectivity, albeit with slightly reduced potency. SN37705 is a 2-urea derivative of the screening hit, and retains its good IDO1 selectivity profile, although with reduced activity. SN36796 contains a pendant phenyl ring at the 6-position of the core, which provides for the possibility of improving IDO1 potency and selectivity by incorporation of appropriate phenyl ring substituents. Only a small representative sample of the W-0019482 analogues that we have synthesised have been presented in Table 2 and Figure 6. The synthesis and activity of additional analogues of W-0019482 will be described in a forthcoming manuscript currently in preparation. Future work will use these analogues to compare the activity of dual inhibitors versus combination treatment of an IDO1-selective together with a TDO2-selective inhibitor, for the control of tryptophan metabolism and reversal of immune suppression in tumour-bearing mice. Such studies will provide insights into the complementary roles of IDO1 and TDO2 in tumour-mediated immune suppression and how to leverage and improve the application of tryptophan dioxygenase inhibitors for the treatment of cancers.

## 3. Materials and Methods

### 3.1. Drugs and Reagents

Chemicals and reagents were sourced from various commercial vendors that included Sigma-Aldred (St Louis, MO, USA), Merck (Darmstadt, Germany), Chembridge (San Diego, SA, USA), Maybridge (Cambridge, UK). Known IDO1 inhibitors, NLG919 was purchased from Selleckchem.com, and INCB14943 was obtained from MedChem Express. Screening of the Walter and Eliza Hall Institute of Medical Research (WEHI, Melbourne, Australia) compound library for IDO1 inhibitors was funded through a University of Auckland Biopharma grant, and compounds identified to be IDO1 inhibitory from the screen were supplied by WEHI for follow-up validation studies in-house at the Auckland Cancer Society Research Centre.

### 3.2. Cell lines and Tissue Culture

The murine GL261 glioma line was obtained from the National Cancer Institute (Frederick, MD, USA) cell line repository under a Materials Transfer Agreement. The murine Lewis Lung carcinoma line, originally from the National Cancer Institute, Bethesda, MD, was adapted for tissue culture (LLTC) at Southern Research Institute, Birmingham [41], and made available to the ACSRC. Both wild-type murine lines were maintained in α-MEM (Gibco BRL, Grand Island, NY, USA) supplemented with 10% FCS and antibiotics (100 U/mL penicillin and 100 µg/mL streptomycin) in a humidified incubator, at 37 °C, with an atmosphere of 5% carbon dioxide in air. Melanoma cell lines developed from surgical specimens obtained from New Zealand patients with informed consent [37], are maintained in α-MEM media with 5% FCS and antibiotics and supplemented with insulin (5 µg/mL), transferrin (5 µg/mL), selenium (5 ng/mL)) (ITS) from Gibco, Grand Island, NY, USA. NZM lines are maintained in low oxygen incubator, at 37 °C, with 5% CO_2_. To induce tryptophan dioxygenase expression, murine cancer cell lines were cultured with mouse IFNγ (100 ng/mL), whilst human melanoma lines were cultured with human IFNγ (80 ng/mL) (BD Pharmingen, San Diego, CA, USA) for 72 h. Kynurenine concentrations in the culture supernatants was used as a surrogate marker of tryptophan dioxygenase activity in cells and was measured using a colourimetric assay as previously described [23].

### 3.3. Engineered LLTC and GL261 Cell Lines Constitutively Expressing Tryptophan Dioxygenases

Transfected LLTC cells constitutively expressing human IDO1 (LLTC-hIDO1) were established as previously described [23], by transfecting wild-type LLTC cells with pAd/CMV/V5-DEST with cDNA for human IDO1 (Invitrogen, Carlsbad, CA, USA). LLTC-mIDO1 cells that constitutive express murine IDO1 were established by transfection of wild-type LLTC with cDNA for murine IDO1 (courtesy of Dr Benoit van den Eynde [2]). GL261 cells were similarly transfected to produce a line constitutively expressing human IDO1 (GL261-hIDO1), and a GL261-hTDO2 line expressing the human TDO2 gene (Source BioScience, Nottingham, England). Stable transfectants were maintained in αMEM with 10% FCS and selection antibiotics (2 µM puromycin for lines transfected with human dioxygenase genes, or 5 µg/mL blastocidin for LLTC-mIDO1 cells).

### 3.4. Western Blots of IDO1 in Cell Lines

Cells were harvested and lysed using Pierce^®^ Radioimmunoprecipitation assay (RIPA) buffer (Thermo Scientific, Rockford, IL, USA) containing 1X Halt™ Protease Inhibitor single use cocktail ((100x), Thermo Scientific, Rockford, IL, USA), and protein concentrations were determined using Bicinchoninic acid (BCA) assay (Thermo Scientific, Rockford, IL, USA). Samples (20 ug) were loaded per well of NuPAGE Bis-Tris minigel (Novex, Carlsbad, CA, USA) together with 8 μL Precision Plus Protein™ Kaleidoskope™ (BioRad, Hercules, CA, USA) and 2 μL Magic Mark™ XP Western (Novex, Carlsbad, CA, USA) protein standards. Gels were electrophoresed at 150 volts for 1 h, and proteins were then transferred to a nitrocellulose membrane. Membranes were blocked in 5% non-fat milk powder in TBS-Tween, and then incubated with primary antibodies to respective tryptophan dioxygenases and then with horse radish peroxidase (HRP)-conjugated IgGs of required reactivity to the primary antibodies, diluted to the required working concentrations in 5% non-fat milk powder in TBS-Tween. For detection of mouse IDO1, membranes were incubated with rat anti-mouse IDO1 (catalogue #122402, Biolegend, San Diego, CA, USA) and detected with HRP-conjugated goat anti-rat antibody. For detection of human IDO1, membranes were incubated firstly with rabbit anti-human IDO1 primary antibody (catalogue # HPA023072, Sigma-Prestige–Atlas, Merck KGaA, Darmstadt, Germany), followed by goat anti-rabbit IgG-HRP secondary antibody (Santa Cruz Biotechnology, Dallas, TX, USA). For quantitation of loading of the lanes, membranes were stripped and incubated with goat anti-GAPDH primary antibody (Santa Cruz Biotechnology, Dallas, TX, USA) followed with donkey anti-goat IgG-HRP secondary antibody (Santa Cruz Biotechnology, Dallas, TX, USA). Chemiluminescent substrate was added to the membranes and images were developed using FUJI LAS-4000.

### 3.5. Immunostaining of Cytospots for Dioxygenase Expression

Cells (10^4^ in 100 mL in 50% FCS in PBS) were pelleted onto Superfrost Plus slides in a cytospin centrifuge (5 min at 400 rpm), and then blocked with 5% goat serum in TBS for 30 mins in a humidity chamber, at room temperature. Intracellular human IDO1 was detected using the same polyclonal rabbit anti-human IDO1 primary antibody (catalogue # HPAD23072) as that used for the Western Blots (Section 3.4), followed by Alexa Fluor 594 conjugated goat anti-rabbit IgG (catalogue # A11072, Invitrogen, Eugene, OR, USA). For murine IDO1, the primary antibody used was rat anti-mouse IDO (clone mIDO-48, catalogue # 122402, Biolegend, San Diego, CA, USA), followed by Alexa Fluor 594 conjugated goat anti-rat IgG secondary antibody (catalogue # 405422, Biolegend, San Diego, CA, USA). Human TDO2 was detected using polyclonal rabbit anti-human TDO2 primary antibody (catalogue # PA5-42759, ThermoFisher, Rockford, USA), followed by the same Alexa Fluor 594 conjugated goat anti-rabbit IgG secondary antibody described above. All cytospots were co-stained with DAPI to detect the nuclei of all cells and imaged using a Zeiss Axioplan 2 fluorescence microscope (Carl Zeiss Meditec, Jena, Germany) at 40× magnification.

### 3.6. Cell-Based Assays for Inhibition of IDO1 or TDO2 and Cytotoxicity of Test Compounds

The cell-based assays using wild-type or engineered IDO1-expressing cells (2 × 10^4^ cells per well) were performed in 96-well plates as previously described [23,42] in α-MEM supplemented with foetal bovine serum (10% *v*/*v*), with the final concentration of DMSO used to dissolve the test compounds no higher than 0.5% (*v*/*v*). For the cell-based assays with engineered TDO2-expressing lines (10^4^ cells per well), the α-MEM culture media was additionally supplemented with 800 mM L-tryptophan. In brief, cells were cultured 24 h with the test compounds, and then supernatants from each well were transferred into fresh, flat-bottomed 96-well plates, mixed with trichloroacetic acid (10% final concentration) and incubated for 20 min, at 60 °C. Plates were centrifuged and supernatants transferred and mixed 1:1 with 4-(dimethylamino)benzaldehyde and the absorbance read at 480 nm as previously described [23]. The inhibition of cellular enzyme activity at each drug concentration was calculated as a percent of control wells with no added drug.

The viability of the cells in each well for every experiment was assessed using the 3-(4,5-dimethylthiazol-2-yl)-2,5 diphenyltetrazolium bromide (MTT) colourimetric assay of metabolic activity of the cells. After the removal of supernatant for kynurenine determination, MTT (20 µL at 5 mg/mL) was added per well and the cultures were further incubated until formation of purple formazan crystals was observed. Culture supernatants were removed and DMSO (100 µL per well) was added to dissolve the crystals, and the absorbance at 550 nm of each well was measured immediately with an automated microplate reader (EnSpire, Perkin Elmer, Waltham, MA, USA). Mean viability of cells of triplicate cultures was calculated and expressed as percentage of untreated controls.

### 3.7. Mice and Tumour Implantations

All animal experiments were carried out with University of Auckland Animal Ethics Committee approval (reference # 001190; 0017681 and 001259) and conformed to local institutional guidelines that meet the standards required by the UKCCCR guidelines and in accordance with the declaration of Helsinki. C57Bl/6 mice bred at the Vernon Jansen Unit, University of Auckland were used as the hosts for the growth of GL261 and LLTC tumours. Unless stated, first-generation tumours, initiated by injecting 10^6^ cells in 100 µL medium subcutaneously into the left flank of each mouse were used for experiments. For one study, subsequent generations of LLTC-hIDO1 tumours were obtained by serial transfer of 1 mm^3^ fragments of a first generation LLTC-hIDO1 tumour excised from a donor mouse and inserted into a subcutaneous pocket in the left flank of anaesthetized (ketamine (100 mg/kg)/xylazine (10 mg/kg)) recipient mice. The pocket was closed using a Michel clip. Tumours were measured thrice weekly, and tumour volumes were calculated as 0.52a^2^b, where a and b are the minor and major axes of the tumour.

### 3.8. Immune Cell Infiltrates in Subcutaneous GL261-WT or GL261-hIDO1 Tumours

Subcutaneous GL261 glioma tumours were established by injecting 10^6^ GL261-WT or GL261-hIDO1 cells in 100 mL PBS into the left flank of female C57BL/6 mice. Mice were culled 15 days post inoculation, and tumours were enzymatically digested to obtain single-cell suspensions. For flow cytometric analysis, 5 × 10^6^ cells/tube were stained for surface marker using anti-mouse antibody CD45-PerCP5.5, B220-PE/Cy7, CD3-PE, CD8a-APC/Cy7 all from BioLegend San Diego, CA, USA, and CD4-TxRed from Life Technologies, Eugene, OR, USA. Cells were then fixed, followed by nuclear staining with mouse anti-Foxp3-APC (eBioscience, San Diego, CA, USA). Samples were run on the BD LSR-II Flow Cytometer (Becton Dickinson, San Jose, CA, USA), and data were analysed using FlowJo software.

### 3.9. Determination of Tumour and Plasma Kynurenine and Tryptophan Concentrations

C57Bl/6 mice were inoculated subcutaneously with 10^6^ GL261-WT or GL261-hIDO1 cells in 100 mL PBS into the left flank. Plasma and tumours were collected from euthanised mice at 14, 17 and 21 days post inoculation (n = 6 per group) and analysed for concentration of kynurenine and tryptophan. In brief, 100µL of each plasma sample was mixed with 100µL of potassium phosphate buffer (0.05 M, pH 6) containing 100µM of the internal standard, 3-Nitro-L-Tyrosine (Sigma-Aldred, St Louis, MO, USA). Twenty-five µL of 2 mol/L trichloroacetic acid (Merck) was added to precipitate the proteins, and the samples were then vortexed and centrifuged at 12,000× *g* for 6 min, at room temperature. Standard calibration curves made from serial dilution of tryptophan (Sigma) and L-kynurenine (Sigma) were prepared with the same treatment as the samples. All supernatants collected after centrifugation were transferred into micro sampling vials and analysed for kynurenine and tryptophan as previously described [43] by HPLC (HP Agilent 1200, Agilent Technologies, Walbronn, Germany). Kynurenine and tryptophan concentrations in samples were calculated from the standard calibration curves and the kynurenine to tryptophan ratio (K:T) was calculated by dividing the concentration of kynurenine by the tryptophan concentration.

### 3.10. Inhibition of Tumour Growth by Dioxygenase Inhibitors in Mice

C57Bl/6 mice were inoculated subcutaneously into the left flank with 10^6^ GL261-hIDO1 cells and were randomised into treatment groups 6 days after inoculation when the tumours were approximately 3 × 3 mm in size. Mice with tumours were treated daily with ip injection of the inhibitor at a tolerated non-toxic dose, dissolved to the required concentration in 50 µL DMSO per mouse. Control groups received only DMSO (≤50 µL). Tumour sizes were measured thrice weekly, as described in Section 3.7.

### 3.11. Statistical Analyses

Graphs were plotted in Prism for Windows, version 6.03 (GraphPad Software, La Jolla, CA, USA). Data between untreated and treated groups, including growth of treated and untreated tumours were compared using Student’s t-tests or ANOVA if multiple comparisons were made and were considered significant when *p* ≤ 0.05. Graph for Figure 1D was generated using the package ggplot2 (v 3.3.0) in RStudio (v 1.3.1093) running on R (v 3.5.3) for Windows. The Spearman’s rank correlation coefficient and the probability value was computed using the function cor.test in the R computing environment.

## 4. Conclusions

In this report, we present the development, characterization and use of murine cancer cell lines engineered to express a unique murine or human IDO1 or TDO2, to test the species specificity and enzyme selectivity of the most potent IDO1 inhibitors discovered in our commissioned screening of the WEHI compound library. The most potent hit identified in our screen, W-0019482, inhibited the growth of engineered GL261-hIDO1 tumours implanted subcutaneously in syngeneic mice and reduced the plasma and intratumoural K:T ratios in the mice. Synthetic modification of the original IDO1-selective hit generated analogues with dual IDO1/TDO2 activity, as well as inhibitors selective for either TDO2 or IDO1. Our results establish the versatility of W-0019482 as a lead in giving rise to the three different subclasses of tryptophan dioxygenase inhibitors. Given the intense efforts, both past and current, in the development of novel inhibitors of tryptophan dioxygenases for research and for use in cancer therapy, we suggest that the inhibitors described in this report will provide excellent tools to elucidate the individual roles and interactions between IDO1 and TDO2 in driving cancer-mediated immune suppression.

## Figures and Tables

**Figure 1 pharmaceuticals-15-01090-f001:**
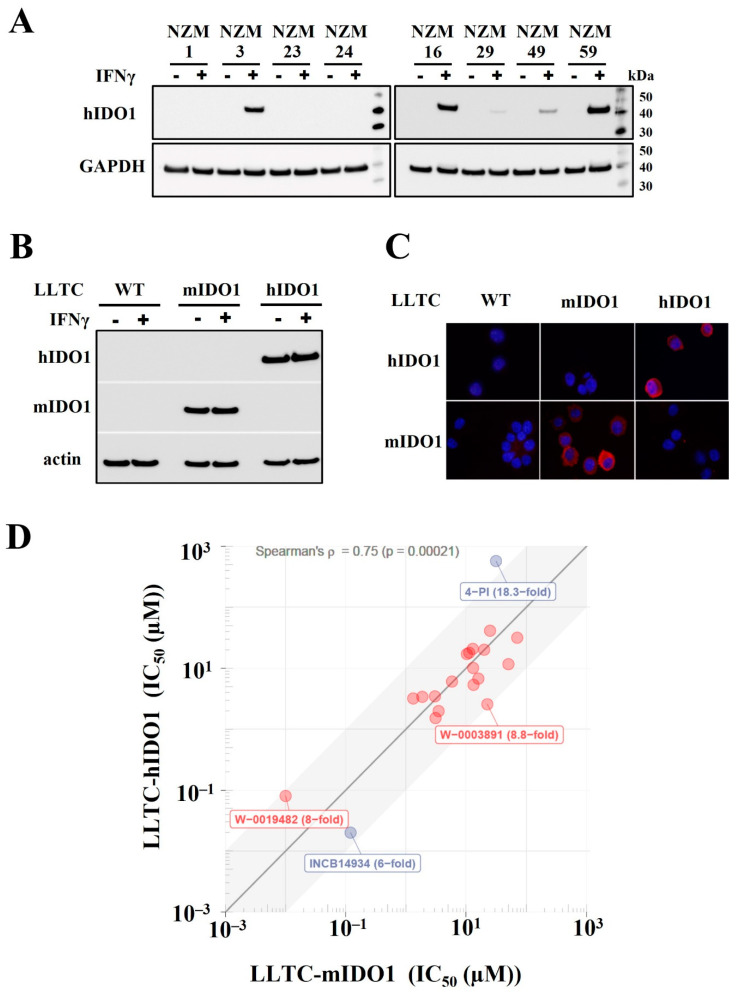
(**A**) Expression of human IDO1 in NZM lines without and after incubation with hIFNγ. (**B**). Expression of murine or human IDO1 in LLTC−WT, LLTC-mIDO1 and LLTC-hIDO1 lines without or after 72 h incubation with mIFNγ. (**C**) LLTC-WT, LLTC-mIDO1 and LLTC-hIDO1 cells immunostained with anti-human IDO1 or anti-mouse IDO1antibodies. (**D**) IC_50_ values measured in cell-based assays against LLTC-hIDO1 plotted against IC_50_ values measured against LLTC-mIDO1 for WEHI compounds (red); reference controls INCB14943 and 4-PI (blue). Spearman’s ρ and the *p*-value in brackets denote Spearman’s rank correlation coefficient and the probability value of that correlation, respectively.

**Figure 2 pharmaceuticals-15-01090-f002:**
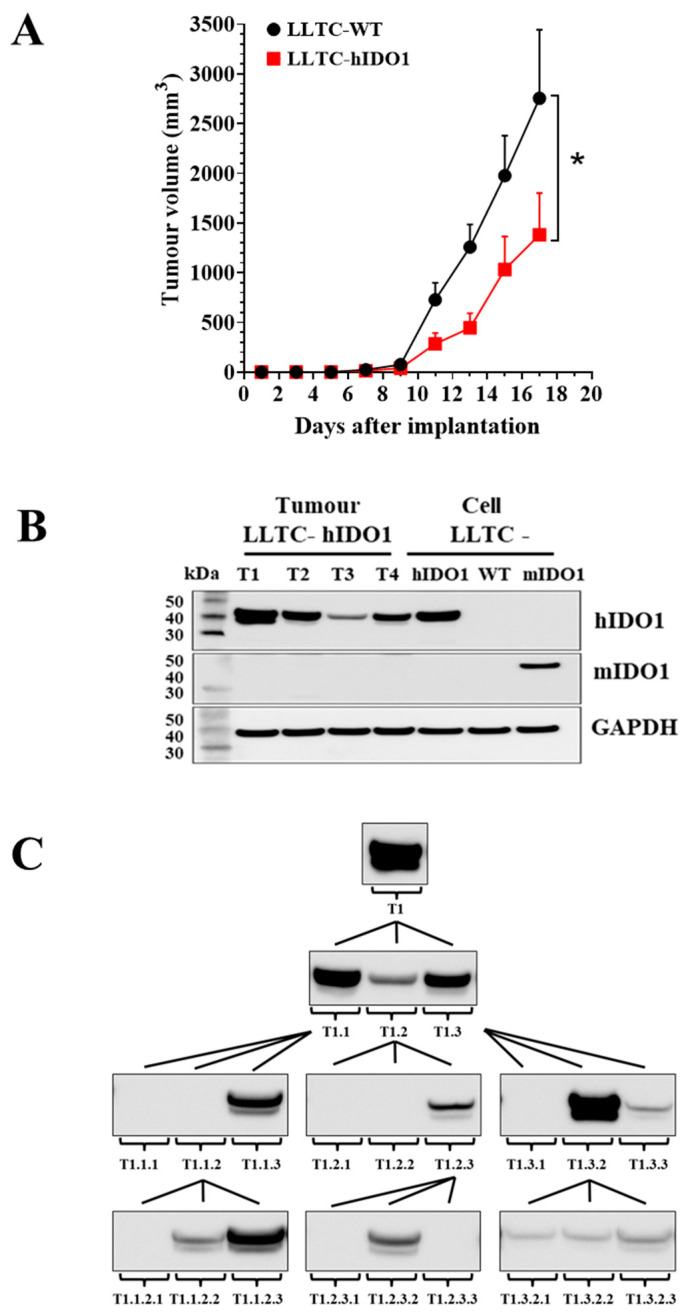
(**A**) Growth of LLTC-WT (black circle) and LLTC-hIDO1 (red square) subcutaneous tumours in C57Bl/6 mice (mean ± s.e.m. of n = 6 per group). *p*-values determined using two-way Anova and * denotes significance *p* ≤ 0.05. (**B**) Western blot of hIDO1 or mIDO1 protein in four individual 14-day LLTC-hIDO1 tumours (T1–T4) and in LLTC-hIDO1, LLTC-WT and LLTC-mIDO1 cells. (**C**) Western blots of hIDO1 expression in individual 22-day tumours from sequential generations of implants from a high-IDO1-expressing LLTC-hIDO1 tumour (T1).

**Figure 3 pharmaceuticals-15-01090-f003:**
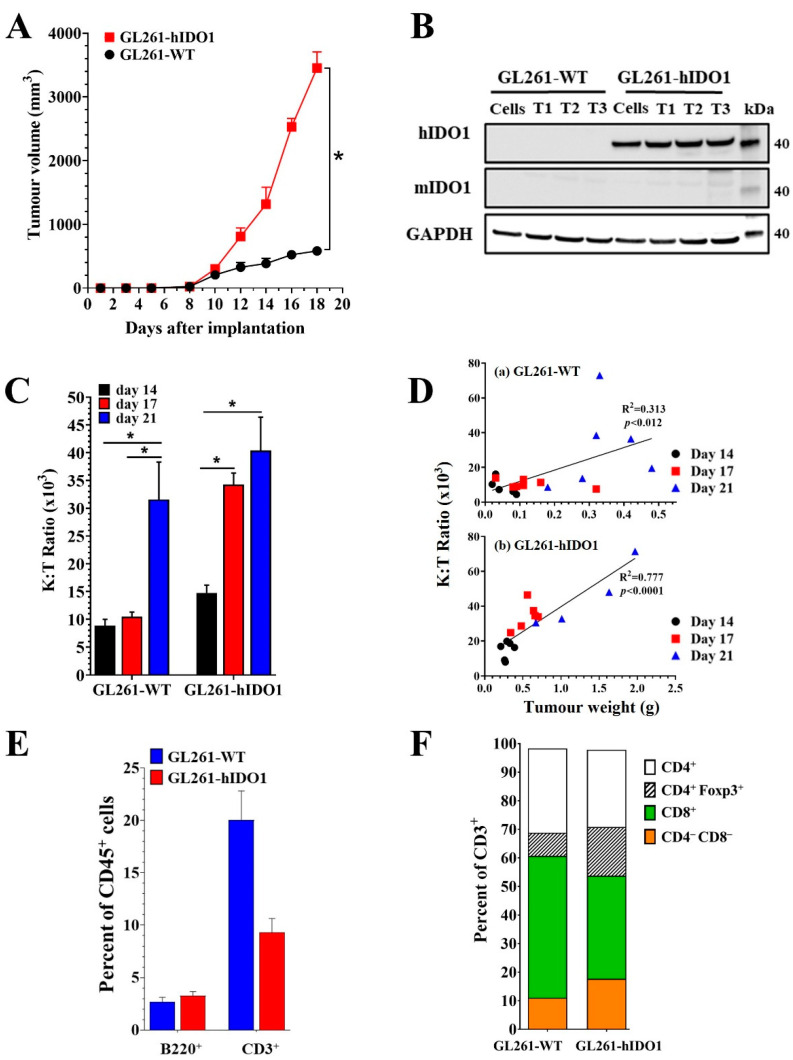
(**A**) Growth of GL261-WT (black circles) and GL261-hIDO1 (red squares) tumours implanted subcutaneously in syngeneic C57Bl/6 mice. *p*-values determined using two-way Anova and * denotes significance *p* ≤ 0.05 (**B**) Western blot of hIDO1 or mIDO1 expression in GL261-WT or GL261-hIDO1 cells or 14-day GL261-WT and GL261-hIDO1 tumours. (**C**) Plasma K:T ratios measured in mice with GL261-WT (a) or GL261-hIDO1 (b) tumours on day 14 (black), day 17 (red) or day 21 (blue) after implantation. (**D**) Plasma K:T ratio plotted against tumour weight and analysed by linear regression and Pearson correlation. (**E**) Percentage of CD45^+^ cells from GL261-WT (blue) and GL261-hIDO1 (red) tumours that were B220^+^ B-lymphocytes or CD3^+^ T-lymphocytes. (**F**) Percent of CD3^+^ T-lymphocytes in GL261-WT or GL261-hIDO1 tumours that were CD4^+^ T-helpers (white), Foxp3^+^ suppressors (hatched), CD8^+^ cytotoxic T-cells (green) and CD4/CD8 null cells (orange). Data are from tumours (n = 3), enzyme digested and the single cells in suspension immunostained with antibodies for the various subtypes and then quantitated by flow cytometry.

**Figure 4 pharmaceuticals-15-01090-f004:**
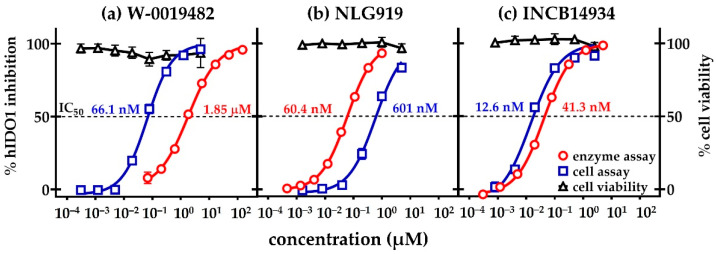
Cell viability (black triangles) and inhibition of hIDO1 activity measured using the LLTC-hIDO1 cell-based assay (blue squares) compared to that measured using an enzymatic assay (red circles) with increasing concentrations of (**a**) W-0019482, (**b**) NLG919 and (**c**) INCB14943.

**Figure 5 pharmaceuticals-15-01090-f005:**
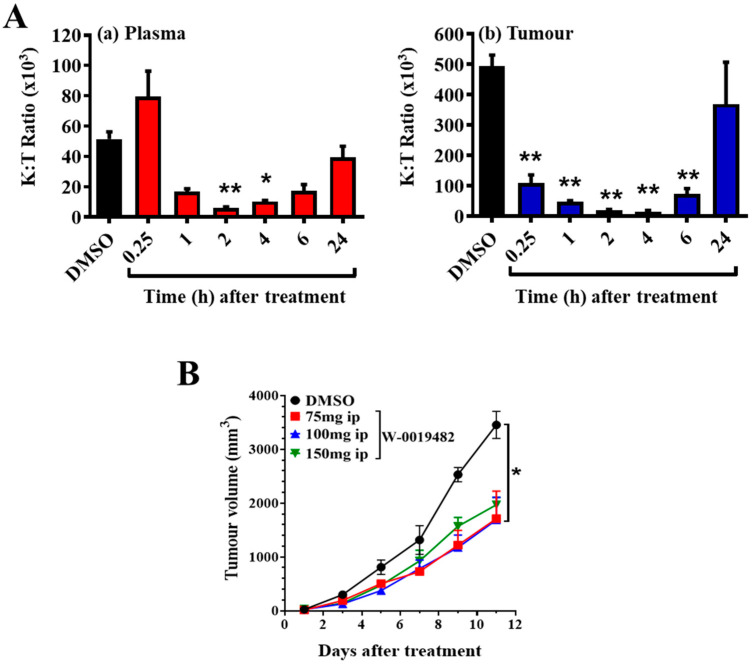
(**A**) K:T ratios at indicated times following treatment with W-0019482 (150 mg/kg) collected from mice with 16-day GL261-hIDO1 tumours (15–20 mm in size), in (a) plasma (red) and (b) tumours (blue). Bars represent mean ± s.e.m. of 3 mice per time point. DMSO vehicle control (black), bar represents mean ± s.e.m. from 21 samples pooled from 3 mice per timepoint following treatment with DMSO. * and ** denote significance (*p* < 0.05, *p* <0.01, respectively) by one-way ANOVA and Sidak’s multiple comparisons compared to DMSO controls. (**B**) Growth of subcutaneous GL261-hIDO1 tumours following daily ip injection of W-0019482 at 75 mg/kg (red squares), 100 mg/kg (blue triangles), and 150 mg/kg (green) compared to DMSO vehicle controls (black). * denotes significant difference (*p* < 0.05) between treated and vehicle controls by two-way ANOVA and Tukey’s multiple comparisons.

**Figure 6 pharmaceuticals-15-01090-f006:**
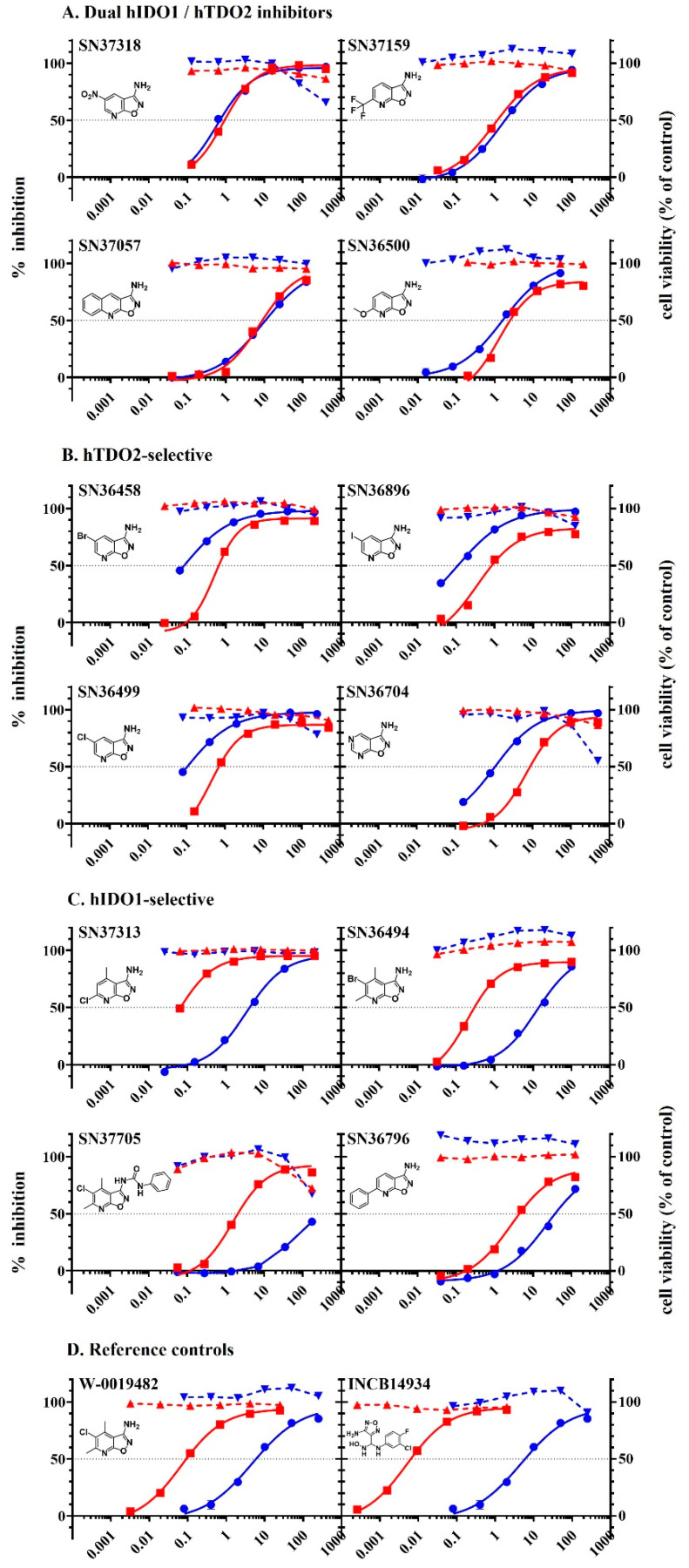
Percent inhibition of hIDO1 (red square) and hTDO2 (blue circle) activity and cell viability (% of control) of LLTC-hIDO1 (red triangle) and GL261-hTDO2 (blue triangle) in cell-based assays versus concentration of inhibitor for (**A**) Dual IDO1/TDO2 inhibitors; (**B**) hTDO2 selective; (**C**) hIDO1 selective; and (**D**) Reference controls W-0019482 and INCB14934.

**Table 1 pharmaceuticals-15-01090-t001:** IC_50_ values for individual WEHI “hits” against LLTC-hIDO1 or LLTC-mIDO1.

Compound	Structure	IC_50_ (µM)
LLTC-hIDO1	LLTC-mIDO1
**W-0019482**	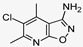	0.08	0.01
**W-0020227**	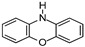	1.52	3.10
**W-0137492**	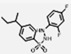	1.98	3.49
**W-0003891**	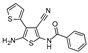	2.56	22.53
**W-0011387**	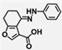	3.19	1.32
**W-0079407**	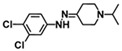	3.39	1.87
**W-0079497**	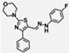	3.45	3.03
**W-0079433**	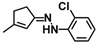	5.33	13.28
**W-0125931**	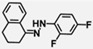	6.04	5.83
**W-0079409**	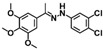	6.74	15.98
**W-0079326**	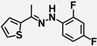	10.11	13.07
**W-0020339**	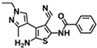	11.64	50.46
**W-0079341**	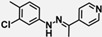	17.08	10.29
**W-0011320**	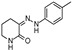	17.95	11.26
**W-0013201**	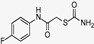	20.09	19.95
**W-0079400**	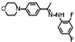	20.77	12.92
**W-0080549**	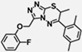	31.61	70.66
**W-0079399**	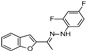	41.23	25.00
**INCB14934**	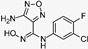	0.02	0.12
**4-PI**	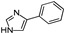	573.50	31.41

**Table 2 pharmaceuticals-15-01090-t002:** IC_50_ values for inhibition in cell-based assays against LLTC-hIDO1 and GL261-hTDO2 of (**A**) dual IDO1/TDO2 inhibitors (**B**) hTDO2-selective and (**C**) hIDO1-selective analogues compared with (**D**) reference controls W-0019482 and INCB14934.

**A. Dual hIDO1 and hTDO2 activity.**
	**Compound**	**Structure**	**IC_50_ (µM)**	**IC_50_ Ratio** **hTDO2/hIDO1**
	**LLTC-hIDO1**	**GL261-hTDO2**
**1**	**SN37318**	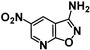	**0.92**	**0.68**	**1.4^−1^**
**2**	**SN37159**	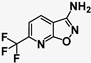	**1.14**	**1.81**	**1.6**
**3**	**SN36500**	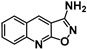	**2.11**	**1.58**	**1.3^−1^**
**4**	**SN37057**	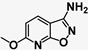	**8.88**	**10.68**	**1.2**
**B. hTDO2 selective.**
	**Compound**	**Structure**	**IC_50_ (µM)**	**IC_50_ Ratio** **hTDO2/hIDO1**
	**LLTC-hIDO1**	**GL261-hTDO2**
**5**	**SN36458**	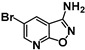	**0.66**	**0.08**	**7.9^−1^**
**6**	**SN36896**	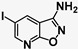	**0.64**	**0.11**	**5.8^−1^**
**7**	**SN36499**	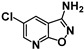	**0.70**	**0.11**	**6.6^−1^**
**8**	**SN36704**	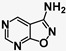	**8.67**	**1.07**	**8.1^−1^**
**C. hIDO1 selective.**
	**Compound**	**Structure**	**IC_50_ (µM)**	**IC_50_ Ratio** **hTDO2/hIDO1**
	**LLTC-hIDO1**	**GL261-hTDO2**
**9**	**SN37313**	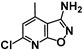	**0.06**	**4.14**	**67**
**10**	**SN36494**	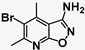	**0.3**	**14.13**	**47**
**11**	**SN37705**	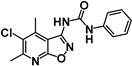	**2.06**	**175**	**85**
**12**	**SN36796**	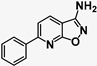	**4.38**	**36.98**	**8**
**D. Reference.**
	**Compound**	**Structure**	**IC_50_ (µM)**	**IC_50_ Ratio** **hTDO2/hIDO1**
	**LLTC-hIDO1**	**GL261-hTDO2**
**Ref 1**	**W-0019482**	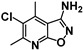	**0.09**	**15.24**	**169**
**Ref 2**	**INCB14934**	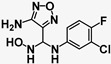	**0.006**	**6.08**	**1013**

## Data Availability

Data is contained within the article and Appendix A.

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
