# Peer review of "Evaluation of Novel Inhibitors of Tryptophan Dioxygenases for Enzyme and Species Selectivity Using Engineered Tumour Cell Lines Expressing Either Murine or Human IDO1 or TDO2"

_pharmaceuticals, 2022, doi:10.3390/ph15091090_

Round 1

Reviewer 1 Report

The article is well reviewed with respect to multiple IDO inhibitors and presents data on immune escape against tumors. The paper is well-written, the tables and figures are of high quality, and the authors have clearly worked hard to produce a comprehensive dataset and detailed description of their methods. This paper is an important contribution and I recommend that it be accepted for publication with minor revision. 

Minor comments:

It would be better if you could provide additional data in Figure 3F. In Figure 3F, the authors show the T-cell fraction, which seems to be the data in peripheral blood. The role of IDO in immune escape would be clearer if the type of T cells around tumor cells could be shown using immunostaining or other methods. The data only from peripheral blood and the percentage of CD3 cells in tumor cells are not convincing enough, and we hope that the data will be revised.

Author Response

We thank reviewer 1 for his/her kind words regarding the quality of our manuscript.

Comment 1. In Figure 3, the authors show the T-cell fraction, which seems to be data in peripheral blood. The role of IDO in immune escape would be clearer if the type of T cells around tumor cells could be shown using immunostaining or other methods.

Response 1: Whilst the K:T ratios were measured in peripheral blood of tumour-bearing mice (Fig 3C), the results in Fig 3E and Fig 3F depict the proportion of various immune cells isolated from day 15 GL261-WT or GL261-hIDO1 tumours (n = 3 for each tumour type). The various subsets of leucocytes in the tumours were quantitated by flow cytometry following immunofluorescence-labelling. Our results clearly show that the IDO1-expressing GL261 tumours contain fewer CD4 cytotoxic T-lymphocytes and a larger proportion of Foxp3+ T-regulatory/suppressor cells compared to the wild-type GL261 tumours (Figure 3F). We used flow cytometric analysis as we believe it provides a more robust, quantitative determination of the number of the various cell types in the whole tumour. Immunostaining of tumour sections will show the distribution of cells from that piece /slice only.

We have altered the Figure legend to clarify that the results in Figure 3E and Figure 3F depict leucocyte subsets in tumours and not peripheral blood.

Reviewer 2 Report

In this manuscript, authors look for potential small molecules to treat cell lines over expressing IDO. Although this is a very interesting approach, authors need to resolve some items before this manuscript can be considered for publication:

1.      It is not clear why the authors decided to use transfected IDO-cells instead of just using a cell line with high basal IDO expression.

2.      Figures need to be fixed. Some of the captions and titles are overlapped.

3.      What is the difference between the tumors generated in figure 1 and 2?

Author Response

Response to Reviewer 2

Point 1: It is not clear why the authors decided to use transfected IDO-cells instead of just using a cell line with high basal IDO expression.

Response 1:  The rationale for our use of transfected lines is provided in Results and Discussion Section 2.1, third paragraph, lines 107-122. We needed a cell line that constitutively expressed human IDO1, suitable for use in vitro screening assays and could also be grown in immune competent mice for subsequent studies of the new inhibitors to reverse IDO1-mediated immune suppression in vivo.

Whilst human cancer cell lines such as the HeLa cells have been widely used by a number of groups for measuring inhibitory activity of compounds against human IDO1 upon stimulation with gamma-interferon in culture, HeLa cells and other human tumour cell lines, cannot be implanted into immunocompetent mice for in vivo studies as they will be rejected by the immune system of the mouse.

We achieved our goal by engineering the murine glioma GL261 line to constitutively express human IDO1, and no murine IDO1 (Figure 3B).  This line grows well after implantation in syngeneic immunocompetent C57Bl/6 mice (Figure 3A), and exhibits reduced killer T-cell number and elevated T-suppressor numbers associated with an immune-suppressed phenotype. This line can be taken forward to evaluate the ability of our novel IDO1 inhibitors to reverse the hIDO1-mediated immune suppression in vivo.

Point 2: Figures need to be fixed. Some of the captions and titles are overlapped.

Response 2: These have been fixed in the re-submission.

Point 3: What is the difference between the tumors generated in figure 1 and 2?

Response 3: There should not be any difference between the three Lewis Lung cell lines used in Figure 1 and those used in Figure 2. An aliquot of the same frozen stock of each of the lines was used to initiate the studies presented in the manuscript. The wild-type Lewis Lung Tissue Culture (LLTC-WT) and the transfected LLTC-hIDO1 and LLTC-mIDO1 cell lines used the in vitro studies in Figures 1B, IC and ID are derived from the same original clones used for inoculation into syngeneic C57Bl/6 immune competent mice to generate subcutaneous tumours for the studies in figure 2.